

# Analysis of the correlation between the serum triglyceride glucose index and the risk of death in patients on maintenance hemodialysis: a retrospective cohort study

Xiaokeng Chi[1,2,3,4,*], Shuxin Chen[4,*], Zhe Huang[5], Rong Zhou[5], Zhicheng Su[4,6], Qiujun Mai[4], Yilin Xu[7] and Jianxin Wan[1,2,3]

[1] Department of Nephrology, Blood Purification Research Center, The First Affiliated Hospital of Fujian Medical University, Fuzhou, Fujian, China
[2] Department of Nephrology, National Regional Medical Center, Binhai Campus of the First Affiliated Hospital, Fujian Medical University, Fuzhou, Fujian, China
[3] Fujian Clinical Research Center for Metabolic Chronic Kidney Disease, Fujian Clinical Research Center for Metabolic Chronic Kidney Disease, Fuzhou, Fujian, China
[4] Department of Nephrology, Blood Purification Research Center, Chaozhou People's Hospital, Chaozhou, Guangdong, China
[5] Clinical Laboratory, Chaozhou People's Hospital, Chaozhou, Guangdong, China
[6] Medical College, Shantou Uinversity, Shantou, Guangdong, China
[7] The Clincal Medical College, Jining Medical University, Jining, Shandong, China
* These authors contributed equally to this work.

Corresponding author
Jianxin Wan, wanjx@fjmu.edu.cn

## ABSTRACT

**Background:** Patients with end-stage renal disease (ESRD) have increased insulin resistance (IR). The triglyceride glucose (TyG) index is a marker of IR and is associated with the prognosis of patients on maintenance hemodialysis (MHD). The aim of this study was to examine the relationship between the TyG index and the risk of death in patients on MHD.

**Methods:** In a retrospective cohort analysis of 368 patients with MHD over an 11-year period from July 1, 2012, to June 30, 2023, the TyG index and other baseline characteristics were measured at the beginning of MHD treatment. A regression model was used to evaluate the correlation between the TyG index and all-cause mortality or cardiovascular mortality in patients on MHD.

**Results:** The TyG index was associated with all-cause mortality and cardiovascular mortality in MHD patients ($P = 0.002$ & $P = 0.015$). After adjusting for various confounding factors, the TyG index remained an independent predictor of all-cause mortality and cardiovascular mortality in MHD patients ($P < 0.001$ & $P = 0.044$). Compared with MHD patients with low TyG index levels, the risk of all-cause mortality and cardiovascular mortality in MHD patients with high TyG index levels increased by 1.790 and 1.735 times, respectively ($P = 0.006$ & $P = 0.022$). The predictive time-AUC values of the TyG index for all-cause and cardiovascular death in MHD patients were between 0.698 to 0.819.

**Conclusion:** The baseline TyG index may be an independent predictor of all-cause mortality and cardiovascular mortality in MHD patients.

# BACKGROUND

Maintenance hemodialysis (MHD) therapy is the most important treatment for end-stage renal disease (ESRD) worldwide (*Himmelfarb et al., 2020*; *Gupta, Woo & Yi, 2021*). A series of factors, such as unequal economic levels, uneven geographical distributions of medical resources, and individual differences in treatment compliance, affect the survival time and quality of life of MHD patients to varying degrees. Cardiovascular disease (CVD) is the leading cause of death in patients with MHD (*Johansen et al., 2023*). The search for effective biomarkers is highly important for the early identification of mortality risk in MHD patients and the development of individualized risk reduction strategies.

Previous studies have shown that parameters related to inflammation, oxidative stress, insulin resistance (IR), and endothelial dysfunction are more predictive of CVD outcome (*Sonmez et al., 2015*). As a new effective evaluation index of IR, the triglyceride glucose (TyG) index is not only directly related to coronary artery disease, myocardial infarction and cardiovascular disease in the general population but also related to all-cause mortality and cardiovascular death in the general population (*Lambrinoudaki et al., 2018*; *Hill et al., 2021*; *Ding et al., 2023*, *2021*; *Xu et al., 2022*; *Liu et al., 2022*; *Lopez-Jaramillo et al., 2023*; *Moon et al., 2023*). Compared with patients with lower TyG indices, patients with higher TyG indices have significantly greater all-cause mortality and cardiovascular mortality (*Xu et al., 2022*; *Liu et al., 2022*; *Lopez-Jaramillo et al., 2023*; *Moon et al., 2023*; *Lertsakulbunlue et al., 2023*). However, to our knowledge, few studies have explored the relationship between IR and CVD in dialysis patients, and the prognostic value of the TyG index in MHD patients is still unclear because impaired renal function affects glucose and lipid metabolism. Therefore, the aim of this study was to explore (1) whether there is a correlation between the baseline TyG index and mortality risk in MHD patients and (2) whether the baseline TyG index has predictive value for all-cause mortality and cardiovascular mortality in MHD patients.

# METHOD

## Participants

This study included patients with MHD who were admitted to the Department of Nephrology of Chaozhou People's Hospital of Guangdong Province between July 2012 and June 2022. The study adhered to all relevant tenets of the Declaration of Helsinki and was approved by the Medical Ethics Committee of the Chaozhou People's Hospital (CZSRMYY-20200317032). Patients whose research has been approved by a medical ethics committee agree to the exemption. The clinical data of all patients were derived entirely from the medical records of the hospital. The inclusion criteria for this study were as follows: (1) aged ≥ 18 years, (2) switched from peritoneal dialysis (PD) to MHD; (3) renal transplantation received prior to undergoing MHD treatment; (4) transferred from other hemodialysis centers after receiving MHD treatment; (5) received regular hemodialysis

treatment less than 3 months; and (6) absence of baseline fasting plasma glucose and triglyceride values. All patients were followed up until June 30, 2023. Finally, a total of 368 patients with MHD were included in this study (Fig. 1).

## Data collection and definitions

All MHD patients were taken as the baseline value measured within one month before the start of the first hemodialysis, and demographic data and clinical data, including gender, age at the first dialysis session, duration of dialysis, comorbidities and clinical testing data, were extracted from the medical information system of Chaozhou People's Hospital.

All MHD patients were rechecked for dialysis-related indicators every 3 months. After fasting overnight before undergoing hemodialysis, 10 ml of fasting venous blood and 5 ml of venous blood at the end of dialysis were collected on the first dialysis day of the week and then sent to the laboratory department of Chaozhou People's Hospital. The routine hematological and biochemical parameters were detected *via* standard laboratory methods, and the corresponding indicators were calculated.

Specifically, the included parameters were as follows: (1) general biochemical index testing: routine indicators of renal function: urea, serum creatinine (SCr), uric acid (UA), fasting plasma glucose (FPG), triglyceride (TG), total cholesterol (TC), hemoglobin (HB), serum albumin (ALB), calcium (Ca), and homocysteine (Hcy); (2) special indicators of renal function: Kt/V (single pool) was calculated *via* the urea kinetic modelling (UKM) formula: $Kt/V = -\ln(R - 0.008T) + (4 - 3.5R) \times UF/W$, wherein R = urea after dialysis/urea before dialysis, T = time of dialysis in hours, UF = ultrafiltration capacity (L), W = postdialysis patient weight (kg); corrected serum calcium (corrected Ca, mmol/L) = measured serum calcium (mmol/L) − [0.025 × albumin (g/L)] + 1; TyG index = ln [fasting serum triglyceride (mg/dL) × fasting plasma glucose (mg/dL)/2] (*Guerrero-Romero et al., 2010*). The above recorded values were the baseline values for 1 month before the start of the patient's first hemodialysis session.

The criteria for hypertension included the following: (1) resting two non-contemporaneous systolic blood pressures ≥ 140 mmHg and/or diastolic blood pressure ≥ 90 mmHg and (2) receiving antihypertensive therapy.

The criteria for diabetes mellitus (DM) include the following factors: (1) patients treated with insulin or oral hypoglycaemic drugs; (2) patients with clinically diagnosed type 1 or type 2 diabetes mellitus; (3) fasting plasma glucose (FPG) ≥ 7.0 mmol/L (126 mg/dL) or 75 g oral glucose tolerance test (OGTT) FPG ≥ 7.0 mmol/L (126 mg/dL) or random blood glucose ≥ 11.1 mmol/L (200 mg/dL) with classic symptoms such as dry mouth, polydipsia, and polyuria; and (4) glycosylated hemoglobin (HbA1c) ≥ 6.5%.

CVD refers to a history of angina, myocardial infarction, arrhythmia, angioplasty, coronary artery bypass grafting, heart failure, or peripheral vascular disease before admission to MHD.

Cardiovascular death was defined as death caused by CVDs such as arrhythmia, cardiac arrest, congestive heart failure, cerebrovascular accidents, ischemic brain injury, hypoxic encephalopathy, peripheral vascular disease, acute myocardial infarction, atherosclerotic heart disease, cardiomyopathy, and aneurysm dissection or rupture.

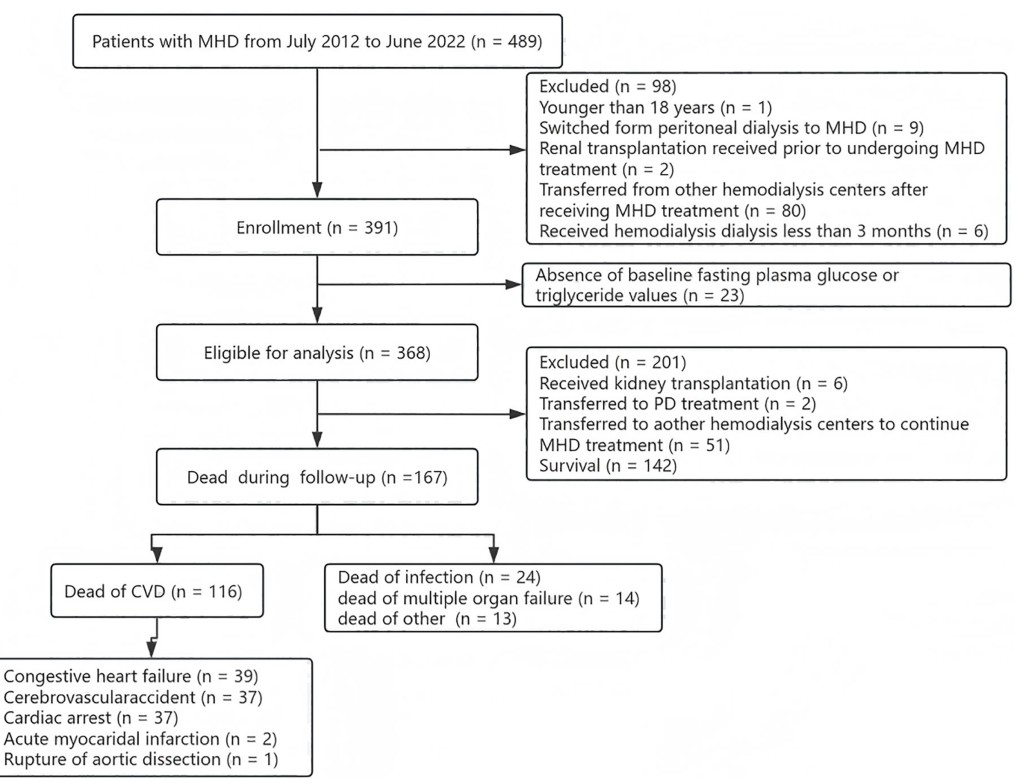

**Figure 1** The study flow chart of the enrollment of MHD patients.

The criteria for hypertriglyceridemia was defined as triglyceride > 1.70 mmol/L.
The criteria for hypercholesteremia was defined as cholesteremia > 5.17 mmol/L.

## Study endpoints

Primary endpoint events: all-cause death and cardiovascular death in patients on MHD.

Patients with MHD who experienced all-cause death were defined as a composite of cardiovascular death, infection, multiorgan failure, and other causes of death.

The follow-up endpoints were as follows: death, transfer to peritoneal dialysis, receipt of kidney transplant therapy, transfer to another blood purification centre for continued treatment, loss to follow-up, or end of the study.

All study endpoints were determined by experts and professors from the Blood Purification Center and hemodialysis nurses.

## Statistical analysis

Continuous variables are expressed as the means ± standard deviations (means ± SDs) or medians ($P_{25}$, $P_{75}$), and the independent samples $t$ test or Mann–Whitney U test was used for comparisons between groups. Categorical variables are expressed as numbers (n) and percentages (%), and comparisons between groups were performed *via* either the chi-square test or Fisher's exact test. Pearson correlation or Spearman correlation was used to analyse the correlation between the TyG index and the clinical parameters of MHD
patients. Kaplan–Meier survival analysis and the log rank test were used to evaluate the survival status between different TyG index groups. Cox regression analysis was used to calculate the risk ratio (HR) and 95% confidence interval (CI) of the risk of death in patients with MHD. Time-dependent area under the curve (Time-AUC) analysis was performed to evaluate the TyG index for predicting death outcome in MHD patients. All the data were statistically analysed *via* SPSS 25.0 and R version 4.4.2. The above tests were all bilateral, and $P < 0.05$ was considered to indicate statistical significance.

# OUTCOME

## Baseline characteristics of MHD patients grouped based on the median TyG index

According to the median TyG index, MHD patients were divided into a low TyG group (TyG < 8.86) and a high TyG group (TyG ≥ 8.86), and the clinical baseline characteristics of the total population and groups were compared (Table 1). The average age of the enrolled MHD patients was 58 (49, 65) years, and 196 were male, accounting for 53.26% of the patients. Compared with the patients in the low TyG group, there were more MHD patients with diabetes mellitus in the high TyG group ($P = 0.017$), and the overall levels of WBC, NE, PLT, FPG, TC, TG and TyG were greater (all $P < 0.05$), but the dialysis adequacy was worse ($P < 0.001$), the dialysis duration was shorter ($P = 0.001$), and there were no obvious differences between the remaining clinical parameters (all $P > 0.05$).

## Correlation analysis between the TyG index and clinical parameters in MHD patients

The TyG index in MHD patients was significantly positively correlated with the WBC, NE, PLT, FPG, TG, TC and Kt/V values (all $P ≤ 0.001$), negatively correlated with K ($r = −0.163$, $P = 0.002$), negatively correlated with ALB ($r = −0.134$, $P = 0.010$), and not significantly correlated with other clinical parameters (all $P > 0.05$) (Table 2).

## Follow-up outcomes of MHD patients

In the 11-year follow-up study, the median survival time of MHD patients was 43 months (ranging from 3 to 132 months), during which 167 (45.38%) all-cause deaths were recorded, 6 (1.63%) received kidney transplantation, 2 (0.54%) were transferred to PD treatment, and 51 (13.86%) were transferred to other blood purification centers for continued dialysis treatment. Among the patients who died of MHD, 116 (69.46%) died of CVD, 24 (14.37%) died of infection, 14 (8.38%) died of multiorgan failure, and 13 (7.78%) died of other causes. Among the 116 MHD patients who died due to CVD, 39 (33.62%) had congestive heart failure, 37 (31.90%) each had cerebrovascular accidents and sudden cardiac death, 2 (1.72%) had acute myocardial infarction, and 1 (0.86%) had aortic dissection rupture.

## Analysis of mortality risk in patients on TyG and MHD

To show the outcomes of MHD patients with different basic TyG indices, we constructed Kaplan–Meier survival plots for all-cause mortality and cardiovascular mortality according

**Table 1 Comparison of the clinical characteristics of MHD patients in different TyG group.**

| | Total population ($n$ = 368) | Low TyG group ($n$ = 179) | High TyG group ($n$ = 189) | P value |
|---|---|---|---|---|
| Gender male | 196 (53.26) | 102 (56.98) | 94 (49.74) | 0.164 |
| Female | 172 (46.74) | 77 (43.02) | 95 (50.26) | |
| Age at first dialysis, year | 58 (49, 65) | 59 (49, 65) | 57 (50, 65) | 0.615 |
| Dialysis duration, months | 44.00 (27.25, 67.00) | 48.00 (31.00, 74.00) | 37 (25.00, 62.00) | 0.001 |
| Complication ($n$, %) | | | | |
| CVD | 294 (79.89) | 145 (81.01) | 149 (78.84) | 0.604 |
| Hypertension | 328 (89.13) | 155 (86.59) | 173 (91.53) | 0.128 |
| Diabetes | 161 (43.75) | 67 (37.43) | 94 (49.74) | 0.017 |
| Others | 72 (19.57) | 39 (21.79) | 33 (17.46) | 0.254 |
| WBC ($\times 10^9$/L) | 7.11 (5.50, 9.54) | 6.61 (5.12, 9.03) | 7.71 (5.76, 10.12) | 0.003 |
| NE ($\times 10^9$/L) | 5.61 (4.23, 7.93) | 5.3 (3.90, 7.31) | 6.00 (4.54, 8.27) | 0.004 |
| LY ($\times 10^9$/L) | 0.67 (0.44, 0.98) | 0.64 (0.42, 0.96) | 0.71 (0.45, 1.03) | 0.143 |
| HB (g/L) | 73.73 (60.25, 86.00) | 70 (59.00, 87.00) | 73.00 (63.00, 85.00) | 0.327 |
| PLT ($\times 10^9$/L) | 184.00 (139.00, 234.75) | 176.00 (132.00, 226.00) | 190.00 (150.00, 251.00) | 0.028 |
| K (mmol/L) | 4.80 ± 1.09 | 4.89 ± 1.12 | 4.72 ± 1.06 | 0.122 |
| Corrected Ca (mmol/L) | 2.06 ± 0.35 | 2.05 ± 0.32 | 2.08 ± 0.37 | 0.293 |
| Urea (mmol/L) | 34.06 (24.88, 43.64) | 32.68 (26.40, 42.51) | 34.25 (23.59, 44.15) | 0.991 |
| SCr (μmol/L) | 1,039.50 (774.3, 1,345.75) | 1,090.00 (770.00, 1,362.00) | 1,000.30 (777.50, 1,311.00) | 0.350 |
| UA (μmol/L) | 552.54 ± 164.36 | 556.43 ± 170.62 | 548.84 ± 158.56 | 0.659 |
| FPG (mmol/L) | 6.36 (5.32, 8.38) | 5.59 (4.79, 6.49) | 7.72 (6.17, 10.43) | <0.001 |
| ALB (g/L) | 33.00 (30.00, 37.00) | 34.00 (30.00, 38.00) | 33.00 (29.00, 36.00) | 0.194 |
| TG (mmol/L) | 1.37 (1.00, 1.82) | 1.04 (0.83, 1.30) | 1.73 (1.44, 2.24) | <0.001 |
| TC (mmol/L) | 4.42 (3.64, 5.35) | 4.08 (3.48, 4.85) | 4.70 (3.94, 5.60) | <0.001 |
| Hcy (μmol/L) | 26.60 (21.23, 32.78) | 26.50 (20.40, 35.40) | 27.75 (21.70, 31.60) | 0.633 |
| Kt/V | 1.24 (1.21, 1.28) | 1.24 (1.23, 1.28) | 1.23 (1.21, 1.27) | <0.001 |
| TyG | 8.86 (8.51, 9.26) | 8.50 (8.22, 8.68) | 9.25 (9.02, 9.58) | <0.001 |

**Note:**

Data are expressed as Mean ± SDs or $n$ (%). $P < 0.05$ was considered statistically significant. WBC, white blood cell; NE, neutrophil; LY, lymphocyte; HB, hemoglobin; PLT, platelet; K, kalium; Ca, calcium; SCr, serum creatinine; UA, uric acid; FPG, fasting plasma glucose; ALB, serum albumin; TG, triglyceride; TC, total cholesterol; Hcy, homocysteine; Kt/V, urea clearance index; TyG, triglyceride glucose index.

to the TyG median grouping (Figs. 2A and 2B). As shown in Fig. 2, MHD patients in the high TyG group had a greater risk of all-cause and cardiovascular mortality than did those in the low TyG group, with a log rank of 13.574 ($P < 0.001$) compared with 15.853 ($P < 0.001$). Although there was no statistically significant difference in all-cause mortality and cardiovascular mortality between the two TyG groups of MHD patients in DM group, MHD patients with high TyG index in diabetic group had a higher risk of death than the same type of non-diabetic patients as shown in Figs. 2C to 2F.

To determine if the increased risk of death among MHD patients in the high TyG group was due to the high prevalence of metabolic syndrome such as diabetes and lipid parameters in this group, we performed subgroup analyses and interaction tests on MHD patients in the high TyG group with or without diabetes, hypertriglyceridemia and hypercholesteremia. As shown in Table 3, the prediction of high TyG index on the

**Table 2 Correlation between the TyG index and clinical variable.**

|  | r | *P* value |
|---|---|---|
| Age at first dialysis, year | 0.039 | 0.450 |
| WBC | 0.208** | <0.001 |
| NE | 0.207** | <0.001 |
| LY | 0.079 | 0.131 |
| HB | 0.087 | 0.094 |
| PLT | 0.174** | 0.001 |
| K | −0.163** | 0.002 |
| Corrected Ca | 0.070 | 0.180 |
| Urea | −0.007 | 0.894 |
| CREA | −0.094 | 0.072 |
| UA | 0.031 | 0.549 |
| FPG | 0.637** | <0.001 |
| ALB | −0.134* | 0.010 |
| TG | 0.767** | <0.001 |
| TC | 0.301** | <0.001 |
| Hcy | −0.025 | 0.635 |
| Kt/V | 0.202** | <0.001 |

Note:
Significant level at ** means two-tailed *P* < 0.01.
* Means two-tailed *P* < 0.05.
WBC, white blood cell; NE, neutrophil; LY, lymphocyte; HB, hemoglobin; PLT, platelet; K, kalium; Ca, calcium; SCr, serum creatinine; UA, uric acid; FPG, fasting plasma glucose; ALB, serum albumin; TG, triglyceride; TC, total cholesterol; Hcy, homocysteine; Kt/V, urea clearance index; TyG, triglyceride glucose index.

all-cause and cardiovascular mortality of MHD patients was not affected by the prevalence of these conditions (all *P* for interaction > 0.05).

## Prognostic value of the TyG index in MHD patients

Cox regression analysis was used to evaluate the predictive value of the TyG index for the mortality risk of MHD patients when it was used as a continuous variable or a categorical variable. The proportional hazards assumptions of TyG index and mortality in MHD patients based on Schoenfeld residuals were shown in Fig. S1–S4. The results showed that the risk proportional assumptions in Cox proportional risk models were congruent (*P* > 0.05). In multivariate Cox regression analysis, three models were developed to evaluate the predictive value of the TyG index for the primary endpoint, including variables significantly associated with the TyG index in Spearman correlation analysis (*P* < 0.05) and/or clinical variables identified as important to death in patients with MHD as confounders. The results of unadjusted Cox regression analysis revealed that the baseline TyG index was a predictor of all-cause and cardiovascular mortality in MHD patients (HR 1.446, 95% CI [1.146–1.824], *P* = 0.002 *vs.* HR 1.443, 95% CI [1.072–1.941], *P* = 0.015). Compared with those in the low TyG group, the risks of all-cause and cardiovascular mortality in MHD patients in the high TyG group were HR 1.771, 95% CI [1.299–2.414], *P* < 0.001 and HR 1.781, 95% CI [1.207–2.556], *P* = 0.003, respectively. After adjusting for gender and age at first dialysis in Model 1, the baseline TyG index was also a predictor of

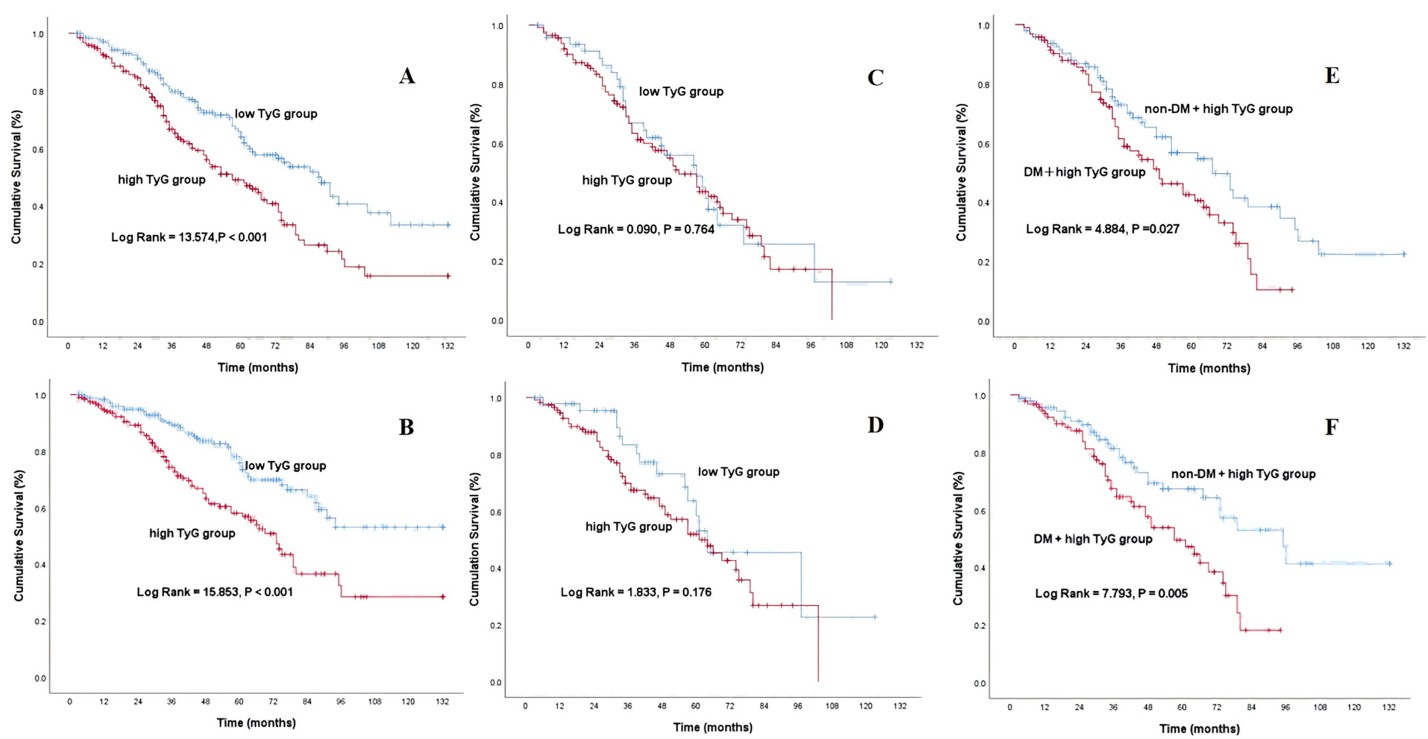

**Figure 2  Kaplan-Meier curve of MHD patients with different TyG index levels.** All-cause mortality (A) and cardiovascular mortality (B) of MHD patients with different TyG index levels; all-cause mortality (C) and cardiovascular mortality (D) of MHD patients with DM in different TyG index groups; all-cause mortality (E) and cardiovascular mortality (F) in MHD patients with or without DM in the high TyG group.

all-cause and cardiovascular mortality in MHD patients (HR 1.429, 95% CI [1.122–1.819], $P = 0.004$ *vs*. HR 1.433, 95% CI [1.065–1.929], $P = 0.017$). Compared with those in the low TyG group, the risks of all-cause mortality and cardiovascular mortality in MHD patients in the high TyG group were HR 1.716, 95% CI [1.258–2.340], $P = 0.001$ and HR 1.773, 95% CI [1.017–2.371], $P = 0.004$, respectively. After Model 1 and adjusting for complications such as CVD, hypertension and DM in Model 2, the baseline TyG index was still a predictor of all-cause and cardiovascular mortality in MHD patients (HR 1.604, 95% CI [1.151–2.237], $P = 0.005$ *vs*. HR 1.582, 95% CI [1.101–2.571], $P = 0.049$). Compared with those in the low TyG group, the risks of all-cause and cardiovascular mortality in MHD patients in the high TyG group were HR 1.557, 95% CI [1.133–2.141], $P = 0.006$ and HR 1.734, 95% CI [1.171–2.219], $P = 0.007$, respectively. Model 3 included Model 2 and WBC, NE, HB, PLT, K, corrected Ca, FPG, ALB, TG, TC and Kt/V, and the baseline TyG index was always an independent predictor of all-cause and cardiovascular mortality in MHD patients (HR 1.550, 95% CI [1.216–1.975], $P < 0.001$ *vs*. HR 1.376, 95% CI [1.009–1.875], $P = 0.044$). Compared with those in the low TyG group, the risks of all-cause and cardiovascular mortality in MHD patients in the high TyG group were HR 1.790, 95% CI [1.177–2.721], $P = 0.006$ and HR 1.735, 95% CI [1.022–2.612], $P = 0.022$, respectively (Table 4).

**Table 3 The subgroup analysis and interaction test of mortality risk for MHD patients in the high TyG group.**

|  | Yes (n, %) | No (n, %) | P value | P value for interaction |
|---|---|---|---|---|
| All-cause death |  |  |  |  |
| DM | 45 (47.4) | 50 (52.6) | 0.984 | 0.904 |
| Non-DM | 53 (56.4) | 41 (43.6) |  |  |
| Hypertriglyceridemia | 57 (58.2) | 41 (41.8) | 0.777 | 0.723 |
| Non-hypertriglyceridemia | 41 (45.1) | 50 (54.9) |  |  |
| Hypercholesteremia | 58 (50.0) | 58 (50.0) | 0.450 | 0.430 |
| Non-hypercholesteremia | 40 (54.8) | 33 (45.3) |  |  |
| Cardiovascular death |  |  |  |  |
| DM | 29 (30.5) | 66 (69.5) | 0.855 | 0.757 |
| Non-DM | 44 (46.8) | 50 (53.2) |  |  |
| Hypertriglyceridemia | 42 (42.9) | 56 (57.1) | 0.292 | 0.272 |
| Non-hypertriglyceridemia | 31 (34.1) | 60 (65.9) |  |  |
| hypercholesteremia | 40 (34.5) | 76 (65.5) | 0.076 | 0.541 |
| non-hypercholesteremia | 33 (45.2) | 40 (54.8) |  |  |

To assess the value of the TyG index on the risk of death in patients with MHD, we generated time-dependent area under the curve (time-AUC) for further analysis. The predictive time-AUC values of the TyG index for all-cause and cardiovascular death in MHD patients were between 0.698 to 0.819. Meanwhile, the overall trends of time-AUC were upward with the extension of follow-up time (Fig. 3 and Table 5).

TyG index, whether as a continuous variable or a nominal variable, can effectively predict the survival time of MHD patients (with a differentiation of 0.752 to 0.806), and improve the prediction ability of the new clinical prediction model for all-cause death and cardiovascular death of MHD patients, with an improvement ratio of at least 10.11% and a maximum of 56.89% (Table 6).

# DISCUSSION

To our knowledge, this is the first study to identify the baseline TyG index as an independent risk factor for mortality risk in patients with MHD. On the basis of an 11-year retrospective study of single-center MHD patients in a city in South China, cardiovascular death was the most common cause of death in MHD patients, and the TyG index may be a prognostic indicator of the mortality risk of MHD patients. Even after adjusting for various confounding factors, the TyG index is still an independent predictor of all-cause and cardiovascular mortality in patients undergoing MHD. Compared with those in MHD patients with low TyG index levels, the risk of all-cause and cardiovascular death in MHD patients with high TyG index levels increased by 1.790 and 1.735 times, respectively. The predictive time-AUC values of the TyG index for all-cause and cardiovascular death in MHD patients were between 0.698 to 0.819.

**Table 4 Cox regression analysis of baseline TyG index and mortality risk in MHD patients.**

| | TyG as a continuous variable[a] | | | TyG as a nominal variable[b] | | |
|---|---|---|---|---|---|---|
| | HR | 95% CI | *P* value | HR | 95% CI | *P* value |
| **All-cause mortality** | | | | | | |
| Unadjusted | 1.446 | [1.146–1.824] | 0.002 | 1.771 | [1.299–2.414] | <0.001 |
| Model 1 | 1.429 | [1.122–1.819] | 0.004 | 1.716 | [1.258–2.340] | 0.001 |
| Model 2 | 1.604 | [1.151–2.237] | 0.005 | 1.557 | [1.133–2.141] | 0.006 |
| Model 3 | 1.550 | [1.216–1.975] | <0.001 | 1.790 | [1.177–2.721] | 0.006 |
| **Cardiovascular mortality** | | | | | | |
| Unadjusted | 1.443 | [1.072–1.941] | 0.015 | 1.781 | [1.207–2.556] | 0.003 |
| Model 1 | 1.433 | [1.065–1.929] | 0.017 | 1.773 | [1.017–2.371] | 0.004 |
| Model 2 | 1.582 | [1.101–2.571] | 0.049 | 1.734 | [1.171–2.219] | 0.007 |
| Model 3 | 1.376 | [1.009–1.875] | 0.044 | 1.735 | [1.022–2.612] | 0.022 |

Notes:
Model 1, adjusted for gender and age at first dialysis; Model 2, Model 1 + complications; Model 3, Model 2 + WBC, NE, Hb, PLT, K, correlated Ca, FPG, Alb, TG, TC and Kt/V. HR, hazard ratio; CI, confidence interval.
[a] HR is the risk ratio for each additional unit of TyG index.
[b] HR is the risk ratio when the low TyG group is used as the control group.

The TyG index is closely related to the risk of diabetes, and the cause may be related to insulin resistance (*Low et al., 2018*; *Lee et al., 2016*; *Navarro-González et al., 2016*). Previous studies have shown that in the general population and those at high risk of metabolism, the TyG index can better predict the occurrence of diabetes mellitus as a new effective predictor of IR than can the use of fasting glucose or glycosylated hemoglobin (*Lopez-Jaramillo et al., 2023*; *Navarro-González et al., 2016*; *Zhang et al., 2017*). The higher the TyG index is, the lower the sensitivity to insulin in the liver, skeletal muscle, and adipose tissue, the more obvious the IR, and the greater the risk of diabetes. In this study, the number of MHD patients in the high TyG group was significantly greater than that in the low TyG group (*P* = 0.017). This result is consistent with those of previous studies. IR is a disease associated with the risk of type 2 diabetes mellitus (T2DM) and CVD. The TyG index is not only helpful in identifying patients at high risk of CVD in asymptomatic CVD patients with in T2DM but also conspicuously associated with arteriosclerosis and coronary artery disease and so on (*Navarro-González et al., 2016*; *Li et al., 2013*; *Zhao et al., 2019*; *da Silva et al., 2019*). Importantly, traditional cardiovascular disease risk factors, such as hypercholesterolemia and obesity, do not fully explain the increased mortality in patients with ESRD. However, the pervasive metabolic abnormalities in patients with MHD, including high FPG levels, high TG levels, and high uric acid levels, may contribute to an increase in CVD-related mortality (*Ikewaki, 2014*). Glucose and lipid metabolism disorders are the primary causes of the pathophysiological development of IR. Both activate the chronic systemic inflammatory response of MHD patients through IR and act on blood vessels to stimulate atheroplaque formation, increase the risk of coronary atherosclerosis, promote CVD progression, and ultimately lead to adverse cardiovascular events (*Li et al., 2022*; *Uhlig, Levey & Sarnak, 2003*; *Jankowski et al., 2021*; *Afsar et al., 2014*). High IR levels are associated not only with a high risk of CVD but also with CVD

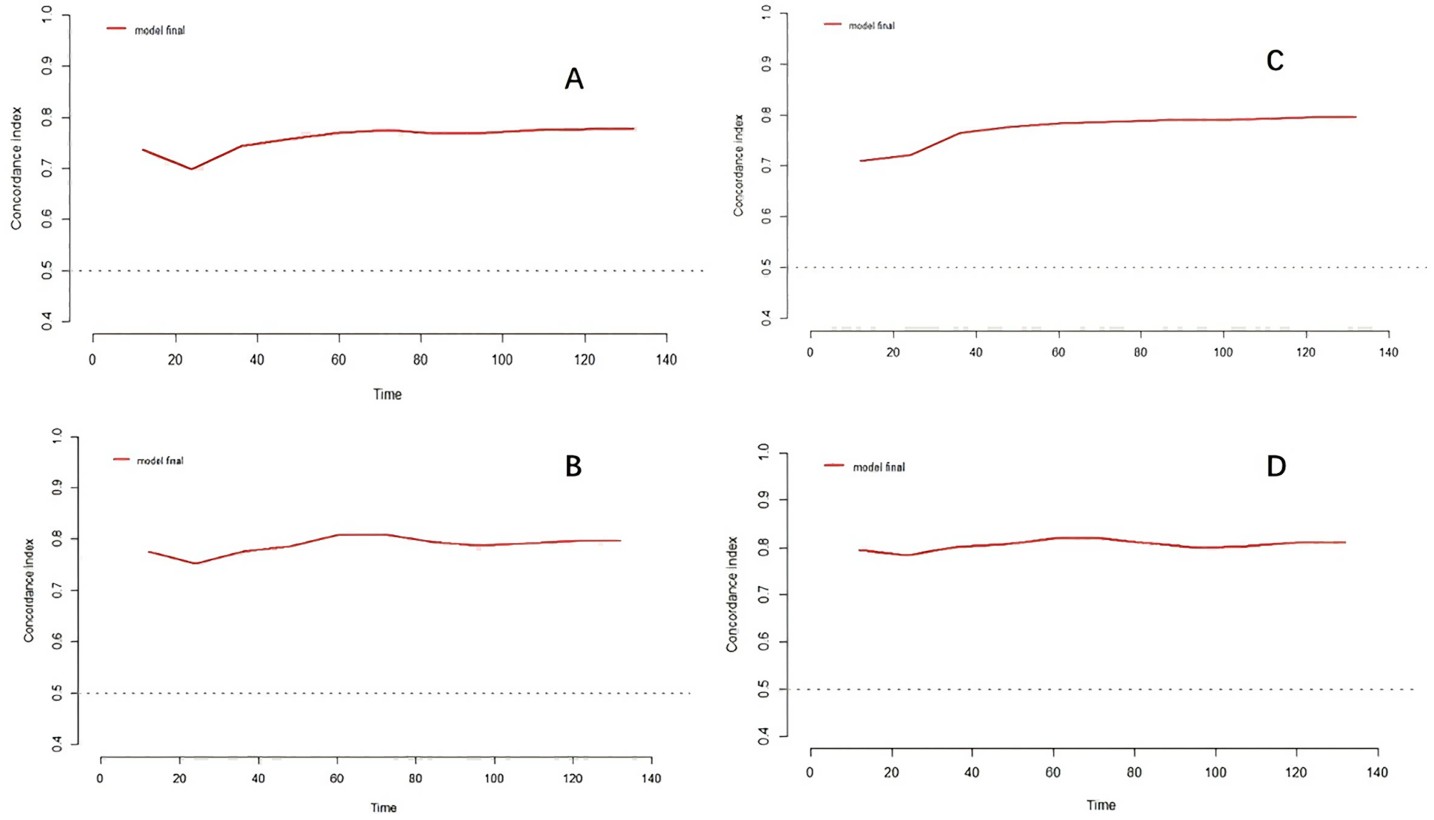

**Figure 3 Time-dependent AUC analysis of TyG index and MHD mortality risk.** The time-dependent AUC between TyG index which as a continuous variable and MHD patients' all-cause death (A) and cardiovascular death (B); the time-dependent AUC between TyG index which as a nominal variable and MHD patients' all-cause death (C) and cardiovascular death (D).     

**Table 5 Time-dependent AUC analysis of TyG index and MHD mortality risk.**

| Time (months) | AUC (%) | | | |
| --- | --- | --- | --- | --- |
| | **All-cause death** | | **Cardiovascular death** | |
| | **TyG as continuous variable** | **TyG as nominal variable** | **TyG as continuous variable** | **TyG as nominal variable** |
| 12 | 73.7 | 70.9 | 77.5 | 79.4 |
| 24 | 69.8 | 72.1 | 75.2 | 78.3 |
| 36 | 74.4 | 76.5 | 77.5 | 80.0 |
| 48 | 75.7 | 77.6 | 78.6 | 80.7 |
| 60 | 77.0 | 78.4 | 80.7 | 81.9 |
| 72 | 77.4 | 78.6 | 80.9 | 81.9 |
| 84 | 76.8 | 79.0 | 79.4 | 80.7 |
| 96 | 76.9 | 78.9 | 78.8 | 79.8 |
| 108 | 77.4 | 79.3 | 79.2 | 80.3 |
| 120 | 77.6 | 79.5 | 79.7 | 80.9 |
| 132 | 77.6 | 79.5 | 79.7 | 80.9 |

**Table 6 Correlation between TyG index and clinical prediction model of mortality risk in MHD patients.**

|  | All-cause death | | Cardiovascular death | |
|---|---|---|---|---|
|  | TyG as continuous variable | TyG as nominal variable | TyG as continuous variable | TyG as nominal variable |
| C_index | 0.752 | 0.769 | 0.787 | 0.806 |
| NRI (%) | 45.65 | 56.89 | 11.38 | 50.20 |
| IDI (%) | 39.34 | 47.24 | 10.11 | 42.67 |

Note:
TyG, triglyceride glucose index; C_index, concordance index; NRI, net reclassification improvement; IDI, integrated discrimination improvement.

mortality (*Adeva-Andany et al., 2019*; *Viigimaa et al., 2020*; *Lambie et al., 2021*). A prospective cohort study in Sweden in a large general population revealed a direct and significant association between the TyG index and all-cause mortality and cardiovascular mortality (*Muhammad et al., 2022*). The results of this study revealed that, compared with those in the low TyG group, the measured levels of metabolic indices (FPG, TG, and TC) in patients receiving MHD with high TyG index levels were obviously increased ($P < 0.05$), but both dialysis adequacy and dialysis duration were remarkably lower ($P < 0.001$), and the number of cardiovascular deaths and all-cause deaths were prominent greater ($P \leq 0.01$). Our findings suggest that a higher TyG index, possibly due to impaired insulin and metabolic status, may be associated with CVD development and even the prognosis of death in patients with MHD. Although there was no statistically difference in the risk of death among MHD patients with DM at different TyG index levels (both $P > 0.05$) in our study, the all-cause and cardiovascular mortality of MHD patients with high TyG index in diabetic group were conspicuously higher than those of non-diabetic patients of the same type (both $P > 0.05$), which further confirmed the above conclusion. Therefore, high TyG index may be an independent predictor of the risk of death in MHD patients diagnosed with DM.

The microinflammatory state is prevalent in patients with ESRD. The microinflammatory state stimulates the body to produce IR through multiple pathways, such as inflammatory factors (WBC, NE, PLT, *etc.*), proinflammatory factors (IL-2, IL-6, TNF-$\alpha$, *etc.*) and acute phase-reactive proteins (ALB, *etc.*). The latter effect on blood vessels to induce endothelial dysfunction and oxidative stress, which worsens renal hemodynamics through various mechanisms, such as sympathetic nervous system and renin–angiotensin–aldosterone system activation, sodium and water retention, and downregulation of the natriuretic peptide system, promotes the progression of renal disease, increases the burden on the circulatory system, increases the possibility of cardiovascular events, and even mortality (*Li et al., 2022*; *Yazıcı & Sezer, 2017*). In this study, the TyG index was not only significantly positively correlated with inflammatory factors (WBC, NE, and PLT) in the microinflammatory state (all $P \leq 0.001$) but also with the levels of inflammatory markers (WBC, NE, and PLT) in MHD patients in the high TyG group, which were obviously greater than those in the low TyG group (all $P \leq 0.05$). In addition, the all-cause and cardiovascular mortality (both $P < 0.001$) of MHD patients in the high TyG index group were noteworthily higher than those in the low TyG index group. These results suggest that high TyG index

levels under the influence of a microinflammatory state may be closely related to the risk of death in MHD patients.

In this study, CVD was the leading cause of death in patients on MHD, which is consistent with the findings of previous studies (*Johansen et al., 2023*; *Fan et al., 2023*). *Cozzolino et al. (2018)* noted that the risk of CVD death in patients on MHD is 20 times greater than that in the general population because of the influence of the mechanism of IR development. IR is closely related to intermediate mechanisms leading to CVD, such as left ventricular hypertrophy, vascular dysfunction, and atherosclerosis, which can significantly predict CVD mortality in ESRD patients (*Lambie et al., 2021*; *Wesołowski et al., 2010*; *Fragoso et al., 2015*). As a surrogate marker of IR, the TyG index is closely related to CVD risk factors and mortality risk. Previous studies have shown strong correlations between the TyG index and arteriosclerosis, coronary artery calcification, and atherosclerotic heart disease (*Lambrinoudaki et al., 2018*; *Hill et al., 2021*; *Ding et al., 2023*; *Ding et al., 2021*). In addition to the TyG index being associated with the incidence of CVD, the higher the TyG index is, the greater the prevalence of symptomatic CVD (*da Silva et al., 2019*; *Sánchez-Íñigo et al., 2016*). In recent years, multiple meta-analyses and longitudinal cohort studies have shown that a higher baseline TyG index is remarkably and independently associated with cardiovascular events, all-cause mortality, and a high risk of cardiovascular mortality in the general population, patients with diabetes, and people without a prior history of CVD (*Xu et al., 2022*; *Liu et al., 2022*; *Lopez-Jaramillo et al., 2023*; *Moon et al., 2023*; *Lertsakulbunlue et al., 2023*). However, the prognostic value of the TyG index for dialysis patients remains known poorly. Even though the study by *Yan et al. (2019)* reported that a higher baseline TyG index in PD patients in China was associated with higher CVD mortality, to date, there have been no reports on the impact of the TyG index on the mortality risk of MHD patients. Although our research was a retrospective evaluation, the patients were based on more real scenarios, and the results revealed that the baseline TyG index was associated with all-cause mortality and cardiovascular mortality. By establishing a clinical prediction model, we found that TyG index could not only accurately predict the survival time of MHD patients during a long follow-up period, but also effectively improve the old model's ability to predict all-cause death and cardiovascular death of MHD patients, with an improvement of up to 56.89%. Meanwhile, this study also confirmed that the high incidence of metabolic syndromes including diabetes, hypertriglyceridemia, and hypercholesterolemia did not affect the ability of high TyG index to predict the risk of death in MHD patients. Adjusting for various confounding factors, the baseline TyG index was always an independent predictor of all-cause mortality and cardiovascular mortality in MHD patients ($P < 0.05$).

There are several strengths to our study. First, our study utilized a cohort design with an extended period of 11 years, as well as relatively comprehensive laboratory data and minimal selection and respondent bias. Second, Chaozhou city is located in the Chaoshan Plain at the border of Guangdong and Fujian Provinces in China. The unique geographical and human characteristics of the area may indicate a representative sample. Third, since a variety of metabolic risk factors may affect the occurrence of IR and CVD, our study measured and analyzed several metabolic factors. Nonetheless, our study is not without

limitations. First, this was a retrospective study, and some unobserved confounders, such as smoking, drinking and others, still remained. Second, since our study measured the TyG index at baseline only, it may not be able to more accurately reflect the influence of dynamic changes in various metabolic risk factors on IR and mortality throughout long-term follow-up. Third, our study focused only on the Chinese population at a single MHD center with a limited sample size, and the applicability of our findings to other geographical regions and different ethnic groups is limited. Therefore, it is necessary to conduct a prospective multi-center study with a large sample size to further explore in the future.

## CONCLUSION

The baseline TyG index, whether as a continuous variable or a nominal variable, was consistently an independent predictor of all-cause mortality and cardiovascular mortality in MHD patients after adjusting for various confounding factors.

### Funding
The authors received no funding for this work.

### Competing Interests
The authors declare that they have no competing interests.

### Author Contributions
- Xiaokeng Chi conceived and designed the experiments, performed the experiments, analyzed the data, prepared figures and/or tables, authored or reviewed drafts of the article, and approved the final draft.
- Shuxin Chen conceived and designed the experiments, performed the experiments, analyzed the data, prepared figures and/or tables, authored or reviewed drafts of the article, and approved the final draft.
- Zhe Huang performed the experiments, analyzed the data, prepared figures and/or tables, and approved the final draft.
- Rong Zhou performed the experiments, analyzed the data, prepared figures and/or tables, and approved the final draft.
- Zhicheng Su conceived and designed the experiments, performed the experiments, analyzed the data, prepared figures and/or tables, and approved the final draft.
- Qiujun Mai performed the experiments, analyzed the data, prepared figures and/or tables, and approved the final draft.
- Yilin Xu performed the experiments, analyzed the data, prepared figures and/or tables, and approved the final draft.
- Jianxin Wan conceived and designed the experiments, prepared figures and/or tables, authored or reviewed drafts of the article, and approved the final draft.

## Human Ethics

The following information was supplied relating to ethical approvals (*i.e.*, approving body and any reference numbers):

Chaozhou People's Hospital granted Ethical approval to carry out the study within its facilities (Ethical Application Ref: CZSRMYY-20200317032).

## Data Availability

data.

## Supplemental Information

Supplemental information for this article can be found online at http://dx.doi.org/10.7717/peerj.18781#supplemental-information.

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
