# Peer review of "Analysis of the correlation between the serum triglyceride glucose index and the risk of death in patients on maintenance hemodialysis: a retrospective cohort study"

_PeerJ, doi:10.7717/peerj.18781_

## Round 0.1 · original submission · Major Revisions

Please address concerns of all the reviewers and amend manuscript accordingly.

Reviewer 1 ·

Basic reporting

The manuscript is well-written and clear, but there are several areas where the English could be improved to ensure technical precision.

1. In the article, urea is referred to using the term BUN (Blood Urea Nitrogen), and the units are given in mmol/L. The authors should clearly state whether they are measuring BUN (just the nitrogen component) or urea (the full molecule). Since the units are in mmol/L, it would be more appropriate to use urea rather than BUN.

2. In Table 3, where it says "TyG as a norminal variable," it should be corrected to "TyG as a nominal variable."

3. The language could be improved by avoiding informal phrasing. For instance, line 308 " ...in diagnosing the risk of death...". Consider professional language editing to ensure clarity.

Experimental design

The research question is well-defined, focusing on whether the TyG index is an independent predictor of all-cause and cardiovascular mortality in patients on maintenance hemodialysis. In populations with conditions like diabetes and metabolic syndrome, the TyG index is recognized as a reliable predictor of CVD and mortality. Research on the TyG index specifically in dialysis patients is not as abundant.

The methodology is described in reasonable detail, but certain aspects could be expanded.

1. The authors use Cox regression models and Kaplan-Meier analysis, but it would be helpful to include more information on how the models were adjusted for specific confounders.
For example, it could clarify the inclusion criteria for variables and the potential multicollinearity checks. Also, given the long follow-up period in your study, it is important to first assess whether the proportional hazards assumption holds (e.g., Schoenfeld residuals in the Cox model).

2. More explanation could be provided on how the ROC curve was used to define optimal cut-off values for the TyG index in the corresponding section (Though the authors labelled Youden index in the Figure 3). As the goal is to find a cut-off point that maximally separates patients based on survival outcomes (all-cause or cardiovascular mortality), the maximally selected rank statistic would be a better choice.

3.Desptie significance, the reported AUC values are only slightly above 0.5, indicating weak discriminatory power. Clinically, a test with such a low AUC may not meaningfully affect treatment decisions or the management of MHD patients. The authors should acknowledge the limited clinical utility of the TyG index in this specific context.

4. To investigate whether the increased risk of death in the high TyG group is due to the higher prevalence of diabetes mellitus in this group, the modifying effect of diabetes (as well as other lipid parameters) should be analyzed. Please add subgroup analysis and interaction test.

Validity of the findings

The study addresses a relevant knowledge gap with appropriate use of statistical methods, a reasonable dataset and an 11-year follow-up. However, while statistically significant, the clinical relevance of the TyG index as a predictor of mortality is limited, and this should be clearly communicated.
The study represents an important step in exploring the TyG index’s potential role in a population where it has not been extensively studied, and the results can serve as a foundation for further research and replication.

Reviewer 2 ·

Basic reporting

In the literature review and hypothesis, the authors suggest that the TyG index is linked to insulin resistance. However, since a significant proportion of dialysis patients have diabetes, these patients already face an elevated mortality risk and are more likely to exhibit higher levels of insulin resistance. This raises concerns about confounding factors that may interfere with the TyG index's predictive ability for mortality. Furthermore, since dialysis represents a late stage of chronic disease with various comorbidities, it is uncertain whether the TyG index behaves similarly in dialysis patients as it does in the general population. The authors should provide actual numerical data comparing the TyG index across different groups, including the general population, diabetic patients, and dialysis patients (with and without diabetes), to better clarify the index's role at different stages of disease. Specifically, the study should include a subgroup analysis of dialysis patients based on diabetes status, along with a discussion of these findings.

Experimental design

1. The authors use the area under the ROC curve (AUC) to evaluate the predictive power of the TyG index. However, AUC may not be the most appropriate metric for survival analysis. Time-dependent ROC curves would be better suited to account for survival outcomes over time.
2. Harrell's c-index would be a more suitable metric for assessing predictive power in survival analysis.
3. In complex chronic diseases, relying on a single variable often leads to weaker predictive power, as seen in the AUC values ranging from 0.5 to 0.6. A more relevant approach would be to assess whether adding the TyG index to the model significantly improves the c-index. Metrics such as Net Reclassification Improvement (NRI) and Integrated Discrimination Improvement (IDI) should also be included. These additional analyses would better support the conclusion that the TyG index is an independent predictor of mortality risk in dialysis patients.

Validity of the findings

The flow chart shows that patients who received kidney transplants or transitioned to peritoneal dialysis were excluded from the study. However, in the outcomes section, the date of kidney transplant or peritoneal dialysis is listed as an endpoint. This discrepancy could lead to confusion about whether these cases were included in the survival analysis. The authors should clarify whether patients who underwent kidney transplantation or peritoneal dialysis were included in the survival analysis and ensure consistency between the flow chart and the outcome definitions.

Additional comments

Minor Comments:
1. The abbreviations used for variables in Table 1 and Table 2 should be clarified by providing both the full names and abbreviations in the footnotes for better understanding.
2. Please verify whether "Hcy" in Table 1 and Table 2 refers to homocysteine or another variable, such as hematocrit.
3. In Line 248 of the manuscript, the number "17" appears before "Importantly, traditional..."—please confirm if this is a reference formatting error or if it has another intended meaning.

Reviewer 3 ·

Basic reporting

good

Experimental design

well done

Validity of the findings

good

Additional comments

The introduction and purpose of the study is well defined. Methods section includes all the necessary explanations and the statistical analysis is enough to show the effect of triglyceride index on mortality among this patient group.
Results section is clear and well defined. Only I offer small changes about the tables(some letter mistakes).
Discussion and conclusion is enough.
I think that the manuscript will contribute to the literature.

---

## Round 0.2 · accepted · Accept

All concerns of the reviewers were addressed and revised manuscript is acceptable now.

Reviewer 1 ·

Basic reporting

The revised manuscript is well-written.

Experimental design

The statistical methods are now appropriate and sufficient to deliver the study results and support the stated conclusions.

Validity of the findings

The study represents an important step in exploring the TyG index’s potential clinical utility in dialysis population.